# RASSF1A Suppression as a Potential Regulator of Mechano-Pathobiology Associated with Mammographic Density in BRCA Mutation Carriers

**DOI:** 10.3390/cancers13133251

**Published:** 2021-06-29

**Authors:** Gina Reye, Xuan Huang, Kara L. Britt, Christoph Meinert, Tony Blick, Yannan Xu, Konstantin I. Momot, Thomas Lloyd, Jason J. Northey, Erik W. Thompson, Honor J. Hugo

**Affiliations:** 1School of Biomedical Sciences, Queensland University of Technology, Brisbane, QLD 4006, Australia; gina.reye@hdr.qut.edu.au (G.R.); x32.huang@hdr.qut.edu.au (X.H.); blick_tony@yahoo.com.au (T.B.); e2.thompson@qut.edu.au (E.W.T.); 2Translational Research Institute, Woolloongabba, QLD 4102, Australia; 3Peter MacCallum Cancer Centre, Melbourne, VIC 3000, Australia; kara.britt@petermac.org; 4Herston Biofabrication Institute, Metro North Hospital and Health Service, Brisbane, QLD 4029, Australia; christoph.meinert@qut.edu.au; 5Gelomics Pty. Ltd., Brisbane, QLD 4059, Australia; 6Central Analytical Research Facility, Queensland University of Technology, Brisbane, QLD 4000, Australia; y8.xu@qut.edu.au; 7Faculty of Science, School of Chemistry and Physics, Queensland University of Technology, Brisbane, QLD 4000, Australia; k.momot@qut.edu.au; 8Department of Radiology, The Princess Alexandra Hospital, Woollongabba, QLD 4102, Australia; thomas.lloyd@health.qld.gov.au; 9Department of Surgery, University of California San Francisco, San Francisco, CA 94143, USA; jason.northey@ucsf.edu; 10School of Health and Behavioural Sciences, University of the Sunshine Coast, Sippy Downs, QLD 4556, Australia; 11School of Medicine and Dentistry, Griffith University, Birtinya, QLD 4575, Australia

**Keywords:** mammographic density, *RASSF1A*, *BRCA1/2* mutations, stiffness, breast cancer

## Abstract

**Simple Summary:**

High mammographic density (MD) is a significant risk factor for the development of breast cancer, as is inheritance of mutations in *BRCA1* or *BRCA2* tumour suppressor genes. High MD combined with *BRCA1/2* gene mutations synergistically increases breast cancer risk, yet *BRCA1/2* mutations alone or in combination do not increase MD or exacerbate the inherent tissue stiffness that high MD creates. The molecular basis for this additive effect therefore remains unclear. Our data indicate that the combinatory effect of high MD and *BRCA* mutations on breast cancer risk may be a product of repression of the tumour suppressor gene *RASSF1A*, in regions of increased tissue stiffness.

**Abstract:**

High mammographic density (MD) increases breast cancer (BC) risk and creates a stiff tissue environment. BC risk is also increased in *BRCA1/2* gene mutation carriers, which may be in part due to genetic disruption of the tumour suppressor gene Ras association domain family member 1 (*RASSF1A*), a gene that is also directly regulated by tissue stiffness. High MD combined with *BRCA1/2* mutations further increase breast cancer risk, yet *BRCA1/2* mutations alone or in combination do not increase MD. The molecular basis for this additive effect therefore remains unclear. We studied the interplay between MD, stiffness, and *BRCA1/2* mutation status in human mammary tissue obtained after prophylactic mastectomy from women at risk of developing BC. Our results demonstrate that *RASSF1A* expression increased in MCF10DCIS.com cell cultures with matrix stiffness up until ranges corresponding with BiRADs 4 stiffnesses (~16 kPa), but decreased in higher stiffnesses approaching malignancy levels (>50 kPa). Similarly, higher RASSF1A protein was seen in these cells when co-cultivated with high MD tissue in murine biochambers. Conversely, local stiffness, as measured by collagen I versus III abundance, repressed RASSF1A protein expression in *BRCA1,* but not *BRCA2* gene mutated tissues; regional density as measured radiographically repressed RASSF1A in both *BRCA1/2* mutated tissues. The combinatory effect of high MD and *BRCA* mutations on breast cancer risk may be due to *RASSF1A* gene repression in regions of increased tissue stiffness.

## 1. Introduction

High mammographic density (HMD) is a significant risk factor for BC, which is the leading cause of female cancer-associated death in Australia [1]. HMD is common, with 43% of women aged 40–74 having heterogeneously or extremely dense breasts (percent MD > 50%) [2]. 

HMD shares radiological, morphological, biophysical, and molecular properties with activated BC stroma: it is comprised largely of amorphous collagen, and confers increased tissue stiffness [3], such that fibroblasts from both microenvironments switch on the JNK1 stress signaling pathway [4]. In the breast, increased abundance and organization of extracellular matrix (ECM) is associated with HMD [5], including our own HMD vs. low MD (LMD) ‘within breast’ comparative data, which demonstrates that collagen-rich ECM is the most discriminatory feature of HMD [6]. The accumulation of collagen alone promotes breast cancer in mice [7], and the arrangement of collagens (typical of those seen in HMD) has been associated with poor survival in BC patients [8]. We have shown that collagen alignment is instrumental in maintaining MD in human breast tissue cultured ex vivo [9]. Furthermore, MD has been implicated in creating an environment of increased stiffness, which in turn may be a contributing factor to MD-associated breast cancer risk [10]. Similarly, breast cancer tissues are stiff and ECM stiffening promotes breast cancer invasion and aggression, where stiffness is derived from stromal collagen linearization at the invasive front [11]. 

The *BRCA1* and *BRCA2* genes mediate repair of DNA double strand breaks and thus are termed tumour suppressor genes [12]. Approximately 72 and 69% of women who harbor *BRCA1* or *BRCA2* gene mutations, respectively, are likely to develop breast cancer by the age of 80, equating to a 5–20 fold increased risk of developing breast cancer due to the loss of genetic fidelity in other tumour suppressor genes [13,14]. The identification of these tumour suppressor genes that are targeted in early in the development of breast cancer is of importance in determining which women are at most risk. *RASSF1A* is a tumour suppressor gene, defined by its role in inhibiting cyclin D1 during G1-S phase progression [15]. *RASSF1A* has been demonstrated to be mutated (A133S polymorphism) in *BRCA1/2* mutated gene carriers with early onset breast cancer [16]. This polymorphism renders RASSF1A unable to perform its normal function, leading to microtubule instability and unregulated progression through the cell cycle [15]. 

The contribution of high MD and BRCA mutation status, combined, to breast cancer risk has been to date, a topic of much interest. It appears that MD is not necessarily raised in female carriers of *BRCA1/2* gene mutations [17,18]. In regards to their combinatory effect, MD and *BRCA1/2* gene mutation carrier status in one study did not report a link to enhanced breast cancer risk [19], whereas in the largest cohort study of women to date on the subject, it was found that MD was an independent risk factor for breast cancer in *BRCA1/2* gene mutation carriers [20]. This was in line with the initial findings of Mitchell and colleagues [18], who also found a synergistic increase in breast cancer risk due to the combination of high MD and *BRCA1/2* gene mutation carrier status. 

*RASSF1A* expression is silenced by methylation in lung cancer, where this silencing was shown enable YAP1 nuclear accumulation and PFHA2 expression to promote collagen alignment, abundance, and stiffness [21]. *RASSF1A* expression is also silenced by methylation in breast cancer [22], yet how it is regulated in at-risk but non-cancerous *BRCA1/2* gene mutated tissues, in combination with high MD, is unknown. This study examines the effect of synthetic and tissue-specific manipulated environments of tissue stiffness on *RASSF1A* expression and analyses expression of this tumour suppressor gene in human mammary tissues in-situ derived from wild-type (WT) or *BRCA1/2* gene mutant carriers. Through these means, we sought to examine the effect of stiffness on *RASSF1A* expression in the context of MD and in *BRCA* WT versus mutant backgrounds in order to better understand the combinatory contribution of high MD and *BRCA1/2* gene mutations to breast cancer risk. 

## 2. Results

### 2.1. MD and Tissue Stiffness

To determine the relationship between MD and breast tissue stiffness, two approaches were employed. The first approach involved subjecting small breast tissue pieces from regions of high and low areas of mammographic density, established radiologically (slice mammogram), to rotational rheometry to determine storage modulus values, followed by the measurement of the same breast tissue pieces using MicroCT to determine % HMD. As shown in Figure 1A, a significant correlation was observed (R^2^ = 0.6062, *p* = 0.0228).

A similar correlation (R^2^ = 0.6083, *p* = 0.0017) was observed between percentage density (as determined by Volpara from whole breast mammography) and stiffness (determined by Boyd algorithm [10]) (Figure 1B). Given that the second approach produced a somewhat stronger correlation, a line of best fit equation was derived from this data (displayed on the graph, Figure 1B) and used to estimate the approximate stiffness ranges relevant for the various MD BiRADs categories (Figure 1C). 

### 2.2. Effect of MD-Relevant Stiffnesses Recapitulated in Three-Dimensions (3D) on MCF10DCIS.com *RASSF1A* Gene and Protein Expression

We investigated the effect of this physiologically relevant stiffness range relating to the various MD categories on *RASSF1A* gene and protein expression in the DCIS.com model of early stage human breast malignancy, using culture on Matrigel (50–150 Pa) and a semi-synthetic extracellular matrix mimic based on photocrosslinkable gelatin (gelatin methacryloyl, or GelMA) (0.5–64 kPa), respectively.

As shown in Figure 2A, *RASSF1A* gene expression in DCIS.com cells varied somewhat across the stiffnesses examined, but with a significant (*p* < 0.01) upward trend in stiffnesses approaching between 50–150 Pa (corresponding to BiRADs 1, which contrasted a significant (*p* < 0.0001) downward trajectory of gene expression when stiffness increase through stiffnesses relevant to BiRADs 4 (~5 kPa) to beyond normal mammary stiffness (50 kPa). This observed downward trajectory of *RASSF1A* expression was evident in the comparison of expression of *RASSF1A* at 50 kPa and cells on tissue culture plastic, albeit cultured in 2D, but of a higher stiffness. 

The pattern of RASSF1A protein expression in the DCIS.com cells at 5, 15, and 50 kPa (Figure 2B) was similar to that of the average *RASSF1A* gene expression in these cells at the same respective stiffnesses (Figure 2A), suggesting that the modulation of RASSF1A expression by stiffness is at the transcriptional level, although at the mRNA level (Figure 1), the suggestion of increase from 5 to 15 KPa is not significant, but the decrease from 5 to 50 kPa is.

RASSF1A protein induction in DCIS.com cells correlated with the intensity of p-HDAX expression, a specific marker for DNA damage repair, suggesting that these stiffnesses evoked considerable stress-induced DNA damage on the DCIS.com cells (Figure 2B). 

### 2.3. RASSF1A Expression in DCIS.com Cells Grown within High versus Low MD Microenvironments Cultivated In Vivo

DCIS.com cells give rise in vivo to comedo-type DCIS-like lesions, some of which progress to invasive breast cancers over time. We have previously shown that these lesions grew more rapidly when cultivated in murine biochambers containing high MD human tissue compared to the addition of low MD into the chambers [23]. Short term in vitro 3D culture of DCIS.com cells in 3D of MD-relevant stiffnesses revealed a biphasic expression pattern of *RASSF1A*. 

We hypothesized that examination of RASSF1A protein expression in the DCIS.com cells within the murine biochambers would provide insight into its expression in malignancies promoted by MD. As shown in Figure 3, RASSF1A protein expression was significantly higher (*p* < 0.0001) in the DCIS.com lesions generated in “high” MD environment (defined radiologically), suggesting the greater stiffness within HMD biochambers drove RASSF1A protein expression in these lesions. 

### 2.4. RASSF1A Protein Expression in Regions of Varied Local Stiffness in Non-Mutated (WT) and BRCA Mutated Patient Tissue

It has been demonstrated that abundance of aligned collagen adjacent to mammary glandular structures positively correlates with increasing MD [5]. Given that RASSF1A appeared to be regulated by stiffness in the DCIS.com cells, and aligned collagen contributes to increased tissue stiffness [5,11], we investigated whether the abundance of this aligned collagen directly associated with RASSF1A protein expression in mammary epithelia. Picrosirius red stained human mammary tissue was viewed under polarized light to determine the index of collagen I (red) signal/collagen III (green) signal in regions immediately adjacent to mammary epithelia. In serial sections, RASSF1A cellular expression was determined in epithelia in matching fields. This data was graphed to examine the relationship between these two variables. 

As shown in Figure 4A, RASSF1A protein abundance was positively correlated (*r* = 0.52, *p* = 0.0160) with the abundance of thicker collagen fibril bundling associated with collagen I, consistent with local stiffness causing increased RASSF1A protein synthesis in adjacent epithelia. 

In contrast, when the same analysis was performed in breast tissue from BRCA1-mutated carriers, the relationship between RASSF1A and collagen I abundance was inverted (*r* = −0.68, *p* = 0.0004), suggesting that local stiffness in these tissues repressed RASSF1A protein expression in adjacent epithelia, rather than induced it. In addition, the gradient of this inverse relationship (m; −0.53) was considerably greater than that observed for non-*BRCA* gene mutated patient tissues (part A; m = +0.19). 

### 2.5. Effect of MD on RASSF1A Expression in WT versus BRCA Mutant Mammary Tissue

Given that local mammary gland stiffness contributed by adjacent stroma had such divergent effects on RASSF1A expression in WT versus *BRCA* mutant mammary tissue, we extended our investigation of stromally-contributed stiffness on RASSF1A expression more broadly. New sampling of regions of high versus low MD were selected in the donor tissues, guided by breast slice mammography, and RASSF1A protein expression within glandular epithelium was quantified. 

No significant difference in RASSF1A protein expression was observed between WT and *BRCA*-mutated tissue when tissue was not separated according to density (“RASSF1A ALL” in Figure 5A). However, when density was taken into account, and RASSF1A expression in high over low MD for each respective donor was plotted, a significant (*p* < 0.05) difference was observed (Figure 5A,ii). Although *RASSF1A* gene expression presented in the same way displayed a similar trend, no significant differences were observed (Appendix A). Consistent with the inverse observation in BRCA mutated tissues (Figure 4B), the RASSF1A index in BRCA mutated tissues (Figure 5A,ii) was lower than 1, suggesting that MD dictates RASSF1A protein expression only in combination with *BRCA*1/2 gene mutations. 

## 3. Discussion

Our data collected from DCIS.com cells (WT in regards to *BRCA1/2* gene mutation status), cultured in various 3D cell culture models of extracellular matrix stiffness relevant to MD and within high MD cultivated in mice and in vivo in human breast tissue epithelia, relative to local stiffness measures, have revealed that *RASSF1A* gene and protein is increased with stiffness in these settings. These observations are in direct contrast to our analyses of RASSF1A protein expression in epithelia in *BRCA1/2* gene mutated mammary tissue under mechanical strain: in this setting, RASSF1A was repressed. The findings of this study therefore suggest a molecular basis for the combinatory effect of high MD and *BRCA1/2* mutation on breast cancer risk reported by others [18,20]. A schematic illustrating the pathways proposed can be found in Figure 6. 

Although mammary tissue stiffness has been linked to an increased risk of breast cancer [10], our study is the first of its kind to specifically determine the relationship between human mammary tissue stiffness and MD, and our data suggest that this relationship is linear (Figure 1). We were then able to use this information to determine extracellular matrix stiffness in 3D cell culture, which would be of physiological relevance to MD. Our observed increase in *RASSF1A* gene and protein expression, evident in these DCIS.com cells in response to increasingly stiffer environments, both in vitro (Figure 2) or in situ (Figure 3), and in association with local mammary gland stiffness in WT BRCA patient tissue (Figure 4), may be in order to maintain a differentiated epithelial phenotype (Figure 6, orange arrows and pathway). The hippo signaling pathway is activated when cells within tissues sense an increase in stiffness, where MST1/2 kinases activate LATS/NDR kinases to ultimately phosphorylate YAP/TAZ, which then bind TEAD proteins to allow direct binding to DNA to influence transcription of genes [24]. This pathway is activated to either drive pluripotency or differentiation (Figure 6), a decision that appears to be influenced by the methylation (silencing) status of the *RASSF1A* promoter and complexing of p73 with YAP [25]. When the *RASSF1A* promoter is unmethylated, as is assumed to be the case in MCF10DCIS.com cells and breast tissues in this study from a WT *BRCA* background, and the hippo pathway is activated, YAP binds with p73 to drive genes associated with differentiation. Indeed, in unpublished data from our laboratory, we find a positive correlation between p73 expression and tissue density determined by NMR technology in human mammary tissue, suggesting that in normal tissues this pathway may be in effect. Therefore stiffness in normal breast tissue associated with MD may act to drive *RASSF1A* expression to sequester YAP down the differentiation pathway (Figure 6A).

It could be argued that an increase in stiffness driving RASSF1A expression in the HMD murine biochambers compared with LMD can only merely be implied, as mechanical stiffness was not directly measured, such as by rotational rheometry, in this setting. Further to this, DCIS.com cells were found to be more metastatic when cultivated in HMD versus LMD tissues [23], therefore more cells moving away from the primary site could have actually led to a *reduction* in stiffness in the HMD biochambers. However, HMD biochambers were heavier than LMD and exhibited a stronger luciferase signal, indicating an increase in DCIS.com cells within a restrained space [23]. This is likely to have caused an increase in pressure by the DCIS.com cells in the HMD chambers, and could be the driving factor in the observed increase in RASSF1A protein expression compared with LMD and Matrigel-only biochambers.

*BRCA1/2* gene mutation status is widely known to result in reduced genome integrity, thus greatly increasing risk of breast and ovarian cancer development [26]. Gao and colleagues found RASSF1A function-depleting single-nucleotide polymorphisms A133S in *BRCA1/2* gene mutation carriers with early-onset breast cancer [16]. Of the 9 *BRCA1* or *BRCA2* gene mutated patient carriers, which were examined in this study, a large proportion had experienced a previous breast cancer. It is therefore reasonable to assume that their remaining “normal” breast tissue that was analyzed in this study could have harbored a small proportion of mutated, dysfunctional *RASSF1A* transcripts, thus *marginally* diminishing the expression of *RASSF1A*, possibly in only regional areas of the tissue. Our data shows that RASSF1A protein was only reduced in *BRCA* gene mutated tissues with increasing local (Figure 4B) or regional (Figure 5) stiffness. As described earlier, tissue stiffness triggers the Hippo signaling pathway including YAP, where the complexing of YAP with p73 triggering differentiation is dependent on RASSF1A expression: a reduction in RASSF1A could therefore decouple YAP and p73, and enable YAP to activate the pluripotency pathway [25]. Pluripotency and stemness are mesenchymal traits [27,28]. Another known factor associated with conversion to the mesenchymal phenotype is ZEB1, which together with MUC1-C has been demonstrated to repress *RASSF1A* expression in human cancer cells [29]. Furthermore, although a mechanism was not elucidated, Pankova and colleagues showed that tissue stiffness repressed RASSF1A in the context of pancreatic cancer [21]. Further study is required to define the pathway of RASSF1A repression occurring in *BRCA1/2* gene mutated tissues of increasing stiffness, which may have direct relevance to understanding pre-malignant changes in the breast in these women who also have high MD.

The local effects of stiffness leading to an inverse in RASSF1A expression (Figure 4B) were specific to *BRCA1* gene mutation carriers only, whereas the more broader effects of stiffness when comparing HMD versus LMD (Figure 5) were observed in both *BRCA1* and *2* gene mutation carriers. The reason for this observation is unclear, however *BRCA2* gene mutation carriers are at a reduced overall breast cancer risk than *BRCA1* gene mutation carriers (both alone and in combination with high MD), as shown in data generated from the Tyrer-Cuzik risk assessment calculator (Appendix A). 

This suggests that potentially the local collagen I/III stiffness effects on RASSF1A, shown in (Figure 4B) to only be divergent (inverse) in *BRCA1* gene mutated tissues, are more potent in evoking tumorigenesis than regional stiffness effects (Figure 5). 

*RASSF1A* is a gene that is frequently silenced by methylation in several cancers including breast [30,31,32]. It is reasonable to speculate that sustained pressure caused by stromally-derived stiffness found in benign breast disease or in malignant states, and exacerbated by *BRCA1/2* gene mutations as depicted in Figure 6, could enable permanent gene silencing via methylation. Thus the “unknown factor” determining whether tissue stiffness represses or promotes RASSF1A expression is likely to be dictated by genes/proteins lost or gained in the process of oncogenesis, as stiffness represses RASSF1A in pancreatic cancer [21]. Further study is required to define this pathway of RASSF1A repression, which may have direct relevance to understanding pre-malignant changes in the breast. Along these lines, “mechanical memory” has been demonstrated for human mesenchymal stem cells cultured within stiff environments to be reminiscent of bone [33,34]. Long-term 3D culture of DCIS.com cells at increasing stiffnesses, coupled with *RASSF1A* promoter methylation analysis at each passage, could shed light on this presumption, as could the introduction of a *BRCA 1* versus *2* gene mutation via CRISPR-Cas9 technology into DCIS.com cells cultured within this 3D stiffness setting. 

## 4. Material and Methods

### 4.1. Preparation of Human Breast Tissue for Stiffness/Density Measurements

Human breast tissue from prophylactic mastectomies were obtained under Metro South Health and Health Service Ethics number HREC/QPAH/16. Demographics of patients involved in this study are shown in Table 1. Tissues were kept on ice from the time of removal until reaching the lab, except for pathology inking and cutting up into ~1 cm slices, routine screening by palpation, selection of slices surplus to pathology needs, and mammography was performed on the breast slices. Areas of high and low density were determined and indicated on the slice mammogram image by a radiologist. Guided by these annotations, mammary tissue pieces of approximately 0.5–1 cm^3^ were excised from the slices, and submerged in Optimal Cutting Temperature (OCT) compound on a circular metal cryostat chuck and incubated for approximately 10 min at minus 20 degrees. Tissue was then cut with a cryostat to obtain an even surface and thickness of approximately 0.5 cm. This tissue, still adhered to the chuck, was inverted onto a microscope slide and OCT allowed to thaw such that the tissue pieces adhered to the slide. Specimens were then transported on slides on dry ice to the Central Analytical Research Facility (CARF) at the Queensland University of Technology for rotational rheometry. Following these procedures, excess liquid OCT was removed using a lint-free tissue and subjected to MicroCT measurements. 

### 4.2. Rotational Rheometry

Breast tissue pieces were measured using a rotational rheometer (Anton Paar MCR302, Graz, Austria) using a 10 mm diameter parallel plate at 37 degrees. Samples were sandwiched between two sandpaper sheets (Norton waterproof abrasive paper P120) (RS Components, Smithfield, Australia), and sandpaper sheets were firmly glued to the bottom of the parallel plate as well as to the stage. Amplitude sweep tests were performed at frequency of 1 Hz with shear strain changing from 0.01 to 20%. A small normal force (0.1N) was applied to the sample during the measurement in order to better grip the sample. 

### 4.3. MicroCT

Volumetric % MD was determined of the breast tissue pieces following rotational rheometry using MicroCT as previously defined [35].

### 4.4. Determining Stiffness from Volpara Data

The algorithm defined by Norman Boyd [10] was applied to determine breast tissue stiffness from data collected from pre-surgery whole breast mammography for the 13 participants (Volpara). 

### 4.5. Culture of MCF10DCIS.com in 3D 

MCF10DCIS.com (DCIS.com) cells (a breast cancer cell line model of early stage human breast malignancy) were obtained from Robert J. Pauley, Barbara Ann Karmanos Cancer Institute, Detroit, MI, USA [36] and cultured as previously described [23]. Matrigel was used for lower range stiffnesses (50–150 Pa) according to manufacturer’s instructions at predetermined concentrations as previously reported [37]. Visible light-crosslinked extracellular matrix mimics based on gelatin methacryloyl (GelMA, type A, porcine skin, 80% degree of amin functionalization, Gelomics Pty Ltd, Brisbane, QLD, Australia) were used for higher range stiffnesses (0.5–50 kPa), as previously described [38]. In both Matrigel and GelMA, DCIS.com cells were resuspended in these matrices and cultured in this manner for 24–48 h at a density of approximately 10,000 cells per mL. 

### 4.6. Quantitative, Real-Time Polymerase Chain Reaction (QPCR)

Following 24–48 h incubation in 3D culture, RNA was extracted from cells using Trizol and agitation using metal beads in round-bottomed tubes loaded onto a Tissue Lyser II machine (Eppendorf South Pacific Pty. Ltd., Macquarie Park, NSW, Australia). cDNA was synthesized and QPCR performed as previously described [9] using primer sequences shown in Table 2.

### 4.7. Embedding and Immunohistochemistry (IHC)

Cell 3D solid suspensions were fixed overnight in 4% paraformaldehyde then placed within a histology mesh in histology cassettes and processed for embedding in paraffin. Immunohistochemistry on formalin fixed-paraffin embedded cells (or human tissues) was performed using the Ventana Ultra. Details of antibodies used in this study can be found in Table 3. For controls, serial sections were treated the same as for test sections, except that the primary antibody was omitted.

### 4.8. Local Stiffness Determined by Collagen I/Collagen III Ratio

Abundance of collagen I as a means to determine mammary tissue stiffness has been used by others and by us [3,9]. Similarly, we stained mammary tissues with picrosirius red and viewed them under polarized light, and used ImageJ to delineate the image into the red and green channels. Relative intensity was then calculated using ImageJ and expressed as a ratio of red (collagen I) versus green (collagen III). The regions of stroma that were captured in the images and quantified were immediately adjacent to glandular epithelium. The ratio is a measure of collagen I versus III abundance. Given that collagen I is a larger, thicker fiber, and collagen III is of a finer, reticular type, and are condensed in a confined space adjacent to glandular epithelium [39], we inferred that this ratio is a determinant of local stiffness relative to MD, as has been proposed by others [5]. 

### 4.9. Visual Selection of High versus Low Areas of Mammographic Density in Assessment of *RASSF1A* Epithelial Expression

Glandular epithelium in a region of low or high local mammographic density was estimated by visually assessing H&E stained tissue for density (abundance of pink, eosin-positive fibrous tissue) adjacent to glands up to a radius of at least 100 and up to 250 M, as is demonstrated in the PSR images shown in Figure 5. 

## 5. Conclusions

The findings from this current study provide a molecular basis for a combinatory effect of high MD and *BRCA1/2* gene mutations on breast cancer risk, through the repression of *RASSF1A*, a tumour suppressor gene known to be silenced by methylation in breast cancers. Our data indicate that Bi-RADs breast density status should be taken into consideration in all decisions regarding breast cancer surveillance and treatment in women harboring BRCA mutations. 

## Figures and Tables

**Figure 1 cancers-13-03251-f001:**
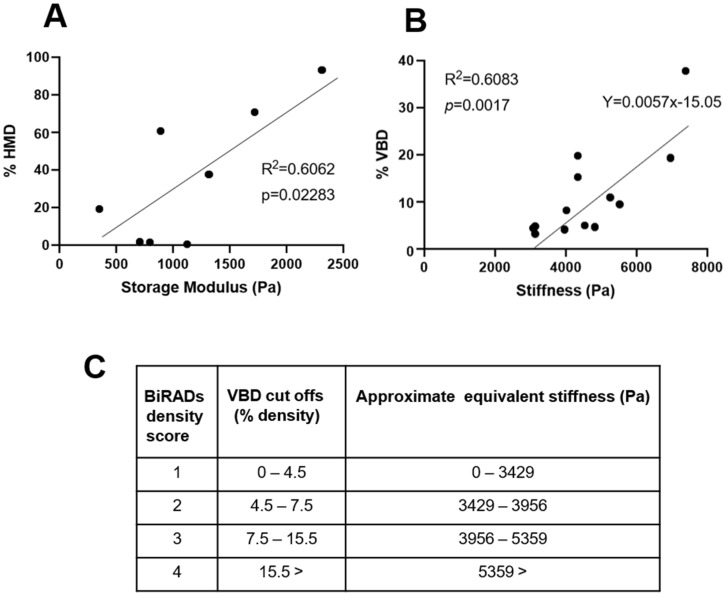
MD positively correlates with breast tissue stiffness. (**A**). Rotational rheometry measured storage moduli of human mammary tissue pieces plotted against % HMD determined by MicroCT. (**B**). Raw DICOM mammogram information and stiffness (n/cm) calculated using the algorithm defined in [10], alongside % density determined using Volpara density software. (**C**). Calculation of equivalent stiffness correlating to BiRADs density score, using equation shown in B. Cut off ranges for BiRADs density categories were obtained from Volpara Breast Density software.

**Figure 2 cancers-13-03251-f002:**
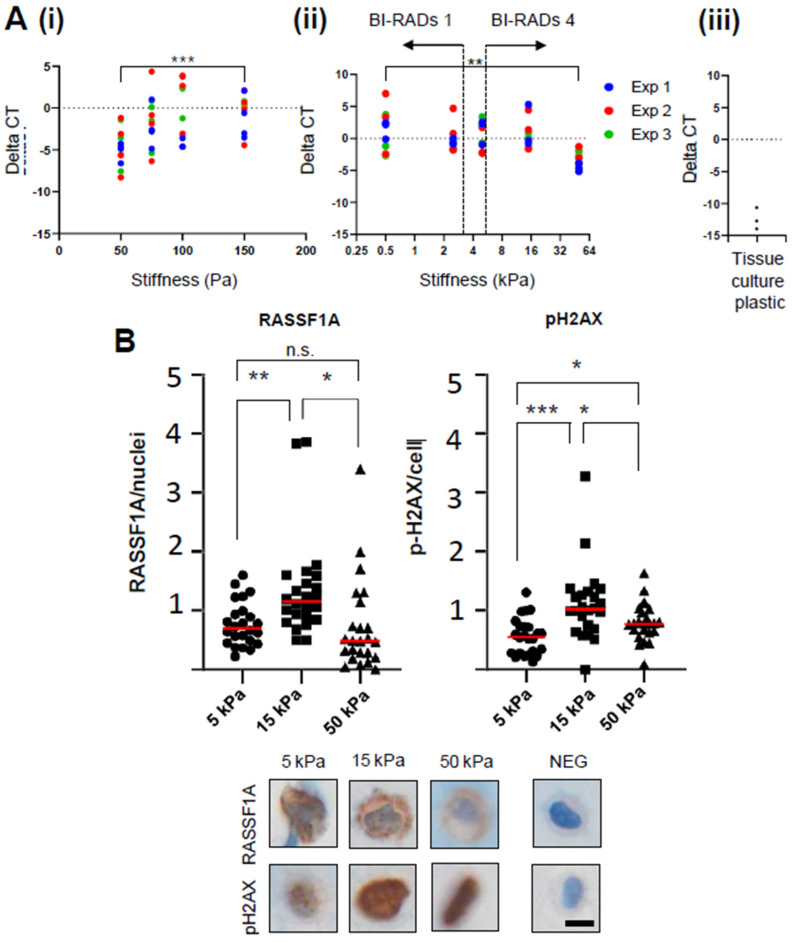
DCIS.com cells grown in 3D matrices of increasing stiffness display a biphasic pattern of RASSF1A gene and protein expression. (**A**). (i) 50–150 Pa achieved with Matrigel; (ii) 0.5–64 kPa achieved with GelMA, (iii) expression in DCIS.com cells grown on tissue culture plastic (>10,000 kPa). Data shown is deltaCT (*RASSF1A* normalized to housekeeping gene *RPL32*) plotted against stiffness. Three independent biological replicates are shown (denoted by color). ** denotes *p* < 0.01, Pearson r correlation test with Gaussian distribution, confidence interval of 95%. Overlaid ranges were derived as depicted in Figure 1C. The red dotted line indicates the upper limit of stiffness of tissues classed as BiRADs 4. BBD = stiffness associated with benign breast disease (between 64 kPa and >10,000 kPa of tissue culture plastic). (**B**) Immunohistochemistry for RASSF1A and pH2AX in formalin fixed, paraffin embedded 3D gel plugs of DCIS.com cells at increasing stiffness. NEG denotes the no-primary antibody control. Representative images are shown, with further images of RASSF1A, pH2AX, and NEG control at 5, 15, or 50 kPa shown in Appendix A, respectively. Cell death was not observed at either stiffness, as shown in representative images in Appendix A. Magnification 40×, scale bar = 5 μm. Student’s t test was used to determine significance. The Shapiro–Wilk test was performed prior to ascertain normality. For all tests used, * denotes *p* < 0.05 and ** denotes *p* < 0.01, *** denotes *p* < 0.001, n.s. denotes no significance.

**Figure 3 cancers-13-03251-f003:**
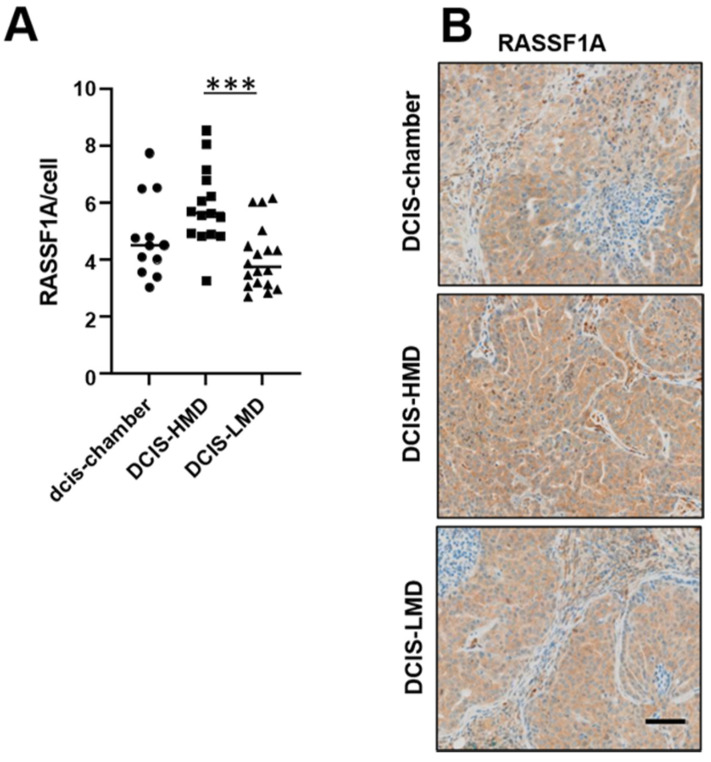
High MD in murine biochambers promotes RASSF1A expression in DCIS-like lesions formed by DICS.com cells. (**A**). Quantitation of RASSF1A intensity per cell; DCIS-chamber *n* = 4, DCIS-HMD *n* = 6, and DCIS-LMD *n* = 6 patient-derived paired (high versus low) biochambers were examined, with 340× magnification fields quantified per patient. *** denotes *p* < 0.001. (**B**). representative images of murine biochamber material, magnification 10×, scale bar = 50 μm. High versus low MD was determined radiologically from the slice mammogram of patient tissue and thus these are relative terms rather than correlative with in vitro data shown in Figure 1.

**Figure 4 cancers-13-03251-f004:**
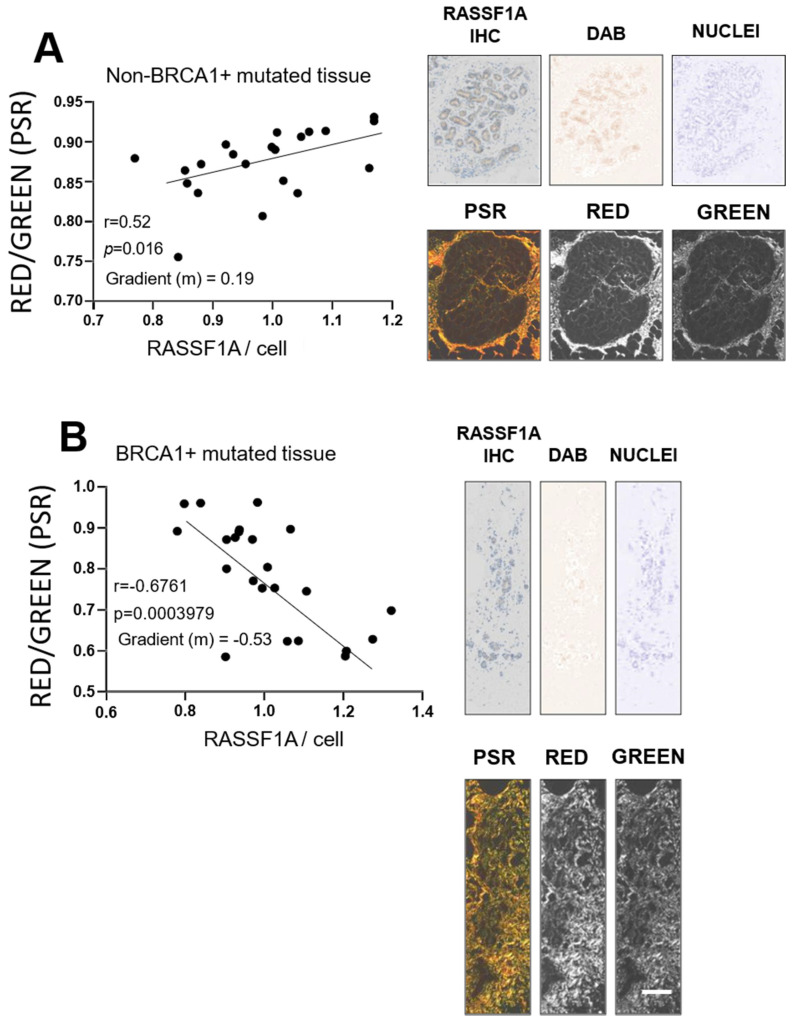
Local stiffness measures, as determined by relative Collagen I vs. III ratio (red vs. green in picrosirius red [PSR] stained tissues viewed under polarized light) versus per-cell RASSF1A expression in stroma adjacent to glandular epithelium non-mutant (**A**) and BRCA1 mutant (**B**) non-malignant human breast tissue. (**A**) A positive correlation was observed for RASSF1A cellular positivity and abundance of thicker collagen fibril bunding associated with Collagen I. (**B**) A negative correlation was observed in patients harboring *BRCA1* gene mutations. Scale bar = 200 μm. 3 images each from at least *n* = 5 patients were examined in A and B. Representative images from patients are shown on right with further images shown in Appendix A.

**Figure 5 cancers-13-03251-f005:**
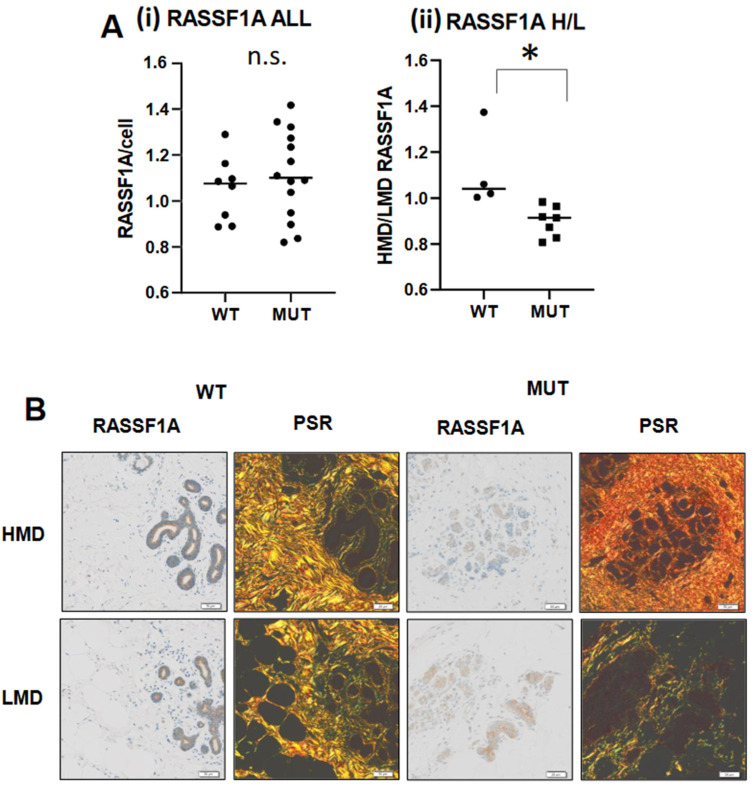
MD influences RASSF1A protein expression in *BRCA1/2* gene mutated tissues. (**A**). RASSF1A protein expression per cell in WT versus *BRCA* mutated tissues (MUT) (i) not separated according to MD; (ii) expression of data in (i) presented as an HMD/LMD index (H/L). Student’s paired *t* test was used to determine significance, where * denotes *p* < 0.05. The Shapiro–Wilk test was performed prior, to ascertain normality. n.s. denotes no significance. (**B**). Representative images of HMD versus LMD in WT versus MUT for RASSF1A and their accompanying PSR images, 10× magnification, scale bar = 50 M.

**Figure 6 cancers-13-03251-f006:**
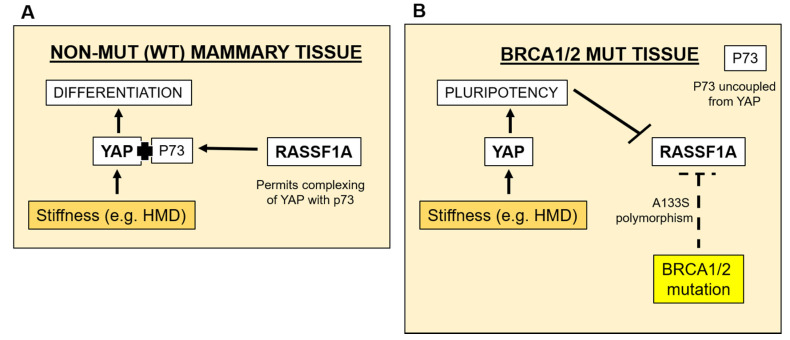
Schematic depicting a possible feedback mechanism through which *RASSF1A* is repressed in pre-malignant mammary tissue not harbouring BRCA mutations (**A**) or harbouring BRCA1/2 mutations (**B**).

**Table 1 cancers-13-03251-t001:** Demographic characteristics of study participants (*n* = 20).

Patient No.	BiRADs Density Status	Age	BRCA Status	Figure
GPH012M	2	53	BRCA1+	Figure 1B, Figure 5
GPH016M	2	46	BRCA2+	Figure 1B
GPH019M	2	43	NEG	Figure 1B
MPRIV015R	1	29	NEG	Figure 1B, Figure 4A, Figure 5
MPRIV017R	2	41	NEG	Figure 1B
PAH009M	4	28	BRCA2+	Figure 1B
PAH021M	1	23	BRCA1+	Figure 1B
PAH023M	2	67	NEG	Figure 1A, Figure 1B
PAH025M	1	34	NEG	Figure 1A
PAH030M	3	47	NEG	Figure 4A, Figure 5
PAH032M	3	48	NEG	Figure 4A, Figure 5
PAH033M	4	41	NEG	Figure 4A
PAH037M	?	27	BRCA1+	Figure 4B, Figure 5
PAH040M	4	48	NEG	Figure 1A, Figure 4A
PAH043M	3	33	NEG	Figure 4A, Figure 5
PAH044M	2	36	BRCA2+	Figure 4A, Figure 5
PAH045M	2	34	BRCA1+	Figure 4B, Figure 5
PAH046M	3	34	BRCA1+	Figure 4B, Figure 5
PAH049M	3	36	BRCA1+	Figure 4B, Figure 5
PAH050M	2	47	RAD51D germline	Figure 5

**Table 2 cancers-13-03251-t002:** Sequences of QPCR primers used in this study.

Primer Name	Forward Sequence (5′-3′)	Reverse Sequence (5′-3′)
RASSF1A	CTCGTCTGCCTGGACTGTTGC	TCAGGTGTCTCCCACTCCACAG
L32	GATCTTGATGCCCAACATTGGTTATG	GCACTTCCAGCTCCTTGACG

**Table 3 cancers-13-03251-t003:** Antibodies used in the current study.

**Antigen**	**Antibody**	**Dilution**	**Supplier**
RASSF1A	Mouse Monoclonal	1:100	ThermoFisher Scientific (Waltham, MA, USA)
Gamma-H2AX (phospho S139)	Rabbit Polyclonal	1:400	Abcam (Cambridge, UK)
**Secondary antigen**	**Antibody**	**Dilution**	**Supplier**
IgG-HRP	Goat Anti-Mouse	1:20,000	Dako, Australia
IgG-HRP	Goat Anti-Rabbit	1:20,000	Dako, Australia

## Data Availability

The data presented in this study are available on request from the corresponding author. The data are not publicly available due to ethical constraints protecting patient privacy.

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
