# Peer review of "RASSF1A Suppression as a Potential Regulator of Mechano-Pathobiology Associated with Mammographic Density in BRCA Mutation Carriers"

_cancers, 2021, doi:10.3390/cancers13133251_

Round 1

Reviewer 1 Report

The manuscript by Reye and colleagues describes a novel role of RASSF1A in the regulation of mechanobiology behavior in BRCA mutation carriers, relating this role with mammographic density. The authors show some in vitro data and ex vivo data that suggest a combination of MD and BRCA mutation, and the RASSF1A downregulation. The study is very interesting. With some clarification, the study provides new information that may be important for understanding the role of RASSF1A in the relationship between MD and BRCA mutations. However, I have some issues with the manuscript detailed below:

MAJOR CONCERNS:

1-Even though, the authors are performing the demonstration in ex vivo tissues and also in one breast cancer cell line, the authors should demonstrate the results in more than one Breast cancer cell line.  This study will be more useful to perform those experimental assays carried out with MCF10DCIS.com, with SUM1315MO2 and SUM149PT that display BRCA mutations (Phenotypic and Molecular Characterization of MCF10DCIS and SUM Breast Cancer Cell Lines, International Journal of Breast Cancer Volume 2013, Article ID 872743).

2-Page 1, Line 22: In the simple summary the authors should add a very brief mention of RASSF1A implication in BRCA mutation and also in stiffness.

3-Line 117: MCF10DCIS is not considered a pre-malignant breast epithelial cell, is considered a pre-invasive breast cancer cell (https://academic.oup.com/jnci/article/92/14/1185a/2905877). The authors should substitute pre-malignant for pre-invasive since a non-invasive BC cell line is unable to form xenografts and to evolve to invasive or metastatic BC, as MCF10DCIS.com does.

4- Conclusions section, Line 382: I am wondering why the authors did not include information about RASSF1A in their conclusion when they focus the title and all the discussion on RASSF1A's role in mechanobiology response in breast cancer tissue.

MINOR CONCERNS:

1-Supplementary images should be also referred to in the paper text, there is only a mention in the text for supplementary figure 4.

2-Line 134 and 135, Figures 2A and 2B (images) display a poor quality, please improve the quality of these figures.

3-Line 163, Figure 3 caption , there is an extra = : DCIS-chamber =n=4

4-Additionally, in supplementary image captions, at least in my word file, the µ symbol appears as another symbol.

5-Line 295, Please try to add the link homogenizing the style and letter size, and reviewing the text.

6-Table 1 should be added as supplementary material

7-Material and Methods section, Line 373: The authors should explain in more detail the relationship between picrosirius red staining and local stiffness.

Reviewer 2 Report

Well written. Pleasant read. Very nice biophysical experiments (tissue stiffness) combined with clinical observations (mammographic density).

The %VBD vs. stiffness correlation (Fig. 1B) is not necessarily linear. I would encourage the authors to explore other statistical fits (power, polynomial, exponential, etc) and determine which type produces the best fit (smallest p-value) and then use that equation. It could be that the linear one is indeed best, but it would be a quick test to verify this.

Figure 2A is fuzzy and there are odd and confusing semi-transparent rectangles?

For the statistical significance analysis, a t-test was used. However, this test assumes a normal distribution of the data. Did the authors test for normality (e.g. Shapiro-Wilk test) prior to using a t-test? If so, then this should be mentioned in the manuscript. If not, then I would recommend doing so. And if the data are not normally distributed, a t-test is no longer valid and an alternative (non-parametric) test should be used (e.g. Wilcoxon ranksum test).

Selecting high vs. low MD guided by slice mammography must come with some uncertainty. Could the authors comment on the extent to which they are confidently working with high or low MD?

Minor remarks in the abstract: MCF10DCIS.com should be briefly described (for a non-expert, is this a website?). Define all acronyms (HMD).

Round 2

Reviewer 1 Report

Thanks for your point by point response